# Early Brain Injury and Neuroprotective Treatment after Aneurysmal Subarachnoid Hemorrhage: A Literature Review

**DOI:** 10.3390/brainsci13071083

**Published:** 2023-07-17

**Authors:** Xiaopeng Li, Lang Zeng, Xuanzhen Lu, Kun Chen, Maling Yu, Baofeng Wang, Min Zhao

**Affiliations:** 1Department of Neurosurgery, Tongji Hospital, Tongji Medical College, Huazhong University of Science and Technology, Wuhan 430030, China; lixiaopeng@tjh.tjmu.edu.cn (X.L.); gdzenglang@163.com (L.Z.); kchen1722@163.com (K.C.); wbf620@163.com (B.W.); 2Department of Neurology, The Third Hospital of Wuhan, Wuhan 430073, China; gdzenglang@hust.edu.cn (X.L.); coney0522@163.com (M.Y.)

**Keywords:** subarachnoid hemorrhage, early brain injury, neuroprotective, cerebral vasospasm

## Abstract

Early brain injury (EBI) subsequent to subarachnoid hemorrhage (SAH) is strongly associated with delayed cerebral ischemia and poor patient prognosis. Based on investigations into the molecular mechanisms underlying EBI, neurovascular dysfunction resulting from SAH can be attributed to a range of pathological processes, such as microvascular alterations in brain tissue, ionic imbalances, blood–brain barrier disruption, immune–inflammatory responses, oxidative stress, and activation of cell death pathways. Research progress presents a variety of promising therapeutic approaches for the preservation of neurological function following SAH, including calcium channel antagonists, endothelin-1 receptor blockers, antiplatelet agents, anti-inflammatory agents, and anti-oxidative stress agents. EBI can be mitigated following SAH through neuroprotective measures. To enhance our comprehension of the relevant molecular pathways involved in brain injury, including brain ischemia–hypoxic injury, neuroimmune inflammation activation, and the activation of various cell-signaling pathways, following SAH, it is essential to investigate the evolution of these multifaceted pathophysiological processes. Facilitating neural repair following a brain injury is critical for improving patient survival rates and quality of life.

Subarachnoid hemorrhage is one of the most dangerous and challenging to treat neurological diseases, particularly aneurysmal SAH (aSAH) [1]. Despite the fact that decades of research have improved our understanding of the pathogenesis of aSAH and the methods for treating ruptured aneurysms, such as surgical clipping or endovascular therapy, aSAH still poses a serious threat to global health, particularly in China [2,3]. The primary cause of impairment and demise in patients with aSAH is the subsequent brain injury [1]. Most in vivo investigations centered on the pathophysiological mechanisms of delayed cerebral vasospasm (DCV) after aSAH or the underlying processes of morphological changes. This is due to the long-standing belief that cerebral arterial spasm, which occurs 3–7 days following aneurysm rupture, is the primary factor contributing to brain injury. However, recent research revealed that patients’ prognoses are not always improved by preventing DCV [4,5]. Recent years have seen a shift in experimental and clinical research attention to the pathophysiologic mechanisms in the first 72 h following a bleeding, also known as “early brain injury” (EBI). EBI was first documented by Kusaka et al. [6] in 2004 and is increasingly acknowledged as a significant factor in delayed cerebral ischemia and long-term morbidity and mortality following SAH [7]. Advancement in early intervention strategies that target the pathogenic mechanisms of EBI would be a vital approach for overcoming the current bottleneck in neuroprotective therapy and research. For the purpose of performing this systematic review, we adhered to the suggested PRISMA guidelines [8]. Furthermore, we recognize the significance of incorporating temporal data to comprehensively analyze the progression of events subsequent to SAH. While our review sheds light on various pathological mechanisms, such as microvascular alterations, ionic imbalances, blood–brain barrier disruption, immune–inflammatory responses, oxidative stress, and cell death pathways, it is essential to investigate their temporal dynamics. By evaluating the sequential evolution of these events, a more comprehensive understanding of early brain injury can be achieved. Moreover, elucidating the temporal course of events holds profound implications for devising effective neuroprotective strategies aimed at specific stages of SAH progression.

## 1. The Mechanism of aSAH-Associated EBI

### 1.1. Alterations in the Brain’s Microcirculation and Energy Metabolism

During the acute phase of SAH, the main factors responsible for the decrease in cerebral blood flow (CBF) are an increase in intracranial pressure (ICP) and a decrease in cerebral perfusion pressure (CPP). Elevated ICP and decreased CPP are caused by hematomas blocking cerebrospinal fluid circulation, cerebrovascular dysfunction causing congestion, and cerebral edema and can be quickly returned to baseline or slightly elevated levels. However, the recovery of CBF takes longer, and the underlying causes may be related to pathophysiological processes, such as microvascular spasm, microthrombus formation, impaired autoregulation, and blood–brain barrier damage [9].

Arterioles are the sites where microvascular spasm mostly happens. When the meningeal vasculature in mice with SAH was examined under a fluorescence microscope, it was discovered that 70% of the vasculature underwent acute contractions (lasting 3–6 h) that might linger up to 72 h, while the veins’ diameters remained unaltered [10]. Small arteries are more likely to be constricted than large arteries, which directly leads to the decrease in CBF and is related to both brain injury and neuronal apoptosis [10]. The SAH investigation in rats indicated that the reactivity of cerebral arterioles to adenosine and nitroprusside was greatly decreased, but the reactivity to CO_2_ was unaffected. This suggests that the vasodilation activity of arterioles is also altered [11,12]. Additionally, 20 min after blood injection, the arterioles’ reactivity to endothelin-1 also increased [13]. Ultrastructural examination of microvessels revealed partially collapsed capillaries, enlarged astrocyte foot processes, and protrusions from endothelial cell lumens. These alterations took place one hour after SAH [14]. The mechanism of microvascular spasm is still unknown, and because of structural variations, it is still uncertain if the findings from studies on big vessel spasm can be directly applicable to microvascular spasm.

Platelets participate in microthrombi development. The primary causes of platelet aggregation and the development of microthrombi are arterial damage and active bleeding. Microvascular platelet aggregation in rats is evident 10 min after SAH, reaches a peak at 24 h, and declines at 48 h [10] and can continue to accumulate in the brain parenchyma [15]. The mechanism underlying platelet aggregation injury involves several key processes. First, mechanical obstruction and biochemical factors, such as the release of serotonin, ADP, and platelet-derived growth factor, contribute to microvessel constriction, leading to reduced cerebral perfusion. Additionally, damage to the vascular endothelium promotes further platelet aggregation. Moreover, the release of collagenases, including MMP-9, can disrupt the blood–brain barrier (BBB), while lymphatic cells adhering to microvessels exacerbate inflammatory damage. Overall, these interconnected processes contribute to the progression of injury caused by platelet aggregation. The peak of microthrombus development in the blood-injection model can be postponed for 48 h. Normal microvessels do not develop microthrombus; only spasmed microvessels do. More microthrombosis results from more severe microvascular spasm, which is directly related to localized brain injury. Therefore, microthrombosis is of great significance to brain injury and prognosis, and the application of u-PA (urokinase-type plasminogen activator) can significantly reduce the formation of microthrombus and reduce the mortality rate [16,17].

Specifically, oxygen and glucose metabolism are two types of brain energy metabolism. Using positron emission tomography (PET), it was shown that the acute phase of SAH is characterized by a drop in cerebral metabolic rate of oxygen (CMRO_2_), which is connected to the reduction in CBF brought on by elevated ICP [18]. Nonetheless, during the initial stage of SAH, CMRO2 decreased more noticeably than CBF, suggesting that we should not only pay attention to the recovery and balance of CBF supply after SAH but also further study the mechanism of brain cell energy metabolism disorder to solve the problem of effective energy utilization of cells. Glucose metabolism is the primary supplier of cellular energy, and the level of glucose metabolism in various regions of the brain can objectively indicate the local functional state of the brain as well as the utilization of glucose by local brain tissue. After SAH, there is an enhancement of high-glucose anaerobic glycolysis, and the more obvious the anaerobic metabolism, the lower the late neurological function score. When patients have neurological abnormalities after a SAH, PET can be utilized to identify the region with impaired glucose metabolism [19,20]. Anaerobic glycolysis compensates for the reduction in aerobic glucose metabolism caused by toxic chemicals after SAH. The energy deficiency cannot be made up for by excessive anaerobic metabolism; instead, it causes energy stress and exacerbates cellular damage. The mechanism of brain energy metabolism deficit following SAH is not well understood; however, signaling pathways, such as the AMPK signaling system, the insulin-mediated signaling network (PI3K/Akt, PI3K/Rac/JNK), and the glucagon-mediated signaling pathway, are all implicated in controlling brain energy metabolism.

### 1.2. Ion Imbalance

After SAH, the microenvironment of brain cells undergoes significant changes in ion balance, including fluctuations in intracellular sodium, potassium, calcium, and magnesium concentrations. Acute short-term increases in cortical potassium ions as well as fluctuations in calcium ions and hypomagnesemia play key roles in these alterations. These dynamic shifts in ion levels have a profound impact on the cellular environment following SAH. All of these alterations can have detrimental effects on damaged brain tissue, including vasoconstriction, interference with nerve electrical activity, and chronic delayed effects through activation of protein expression. The primary result of ion imbalance is cortical spreading depression (CSD) that, together with its secondary cortical spreading cerebral ischemia (CSCI), is now thought to constitute a new etiology of early and late cerebral ischemic injury after SAH. The mechanism is currently a prominent topic in research.

Under the conditions of ischemia, hypoxia, excessive potassium, and other stressors, cortical spreading depolarization (CSD) occurs as a widespread, continuous depolarization of a significant population of neurons in the cortex. The extracellular direct current potential falls, and a significant amount of sodium and calcium influx into the cells; these symptoms spread from the origin to the entire cerebral hemisphere but do not impact the contralateral cerebral hemisphere. CSD can lead to cerebral vasodilation and an increase in CBF in healthy brain tissue [21]. However, in an ischemic and hypoxic environment, CSD results in cerebral vasoconstriction, decreased CBF, and CSCI [22] and interacts with calcium excess, i.e., hyperkalemia, and the discharge of excitatory amino acids can worsen brain damage. The development of CSD after SAH is influenced by oxygenated hemoglobin, increased extracellular potassium, reduced NO availability, glutamate, ET-1 (endothelin-1), and other variables. According to studies, CSD begins shortly after SAH and lasts for a considerable amount of time. Its cluster incidence is strongly correlated with early and late brain injury in both time and space [21].

CSD causes epilepsy, decreased CBF, increased brain metabolism, vasospasm, ion imbalance, and dysfunctional neurovascular coupling. The precise mechanism could be that the high levels of extracellular potassium and excitatory amino acids in SAH cause CSD, and CSD can further cause extracellular potassium ions to rise and excitatory transmitters to be released. Microcirculation dysfunction and a malfunction of neuronal energy metabolism following SAH also cause these conditions. Following SAH, alterations in CBF and metabolism give rise to modifications in the functioning of brain cell membranes, which affect the resting potential and action potential of cells to form CSD, and CSD further affects the cell membrane potential, resulting in epileptic seizures. This vicious cycle further aggravates CBF and oxygen supply disorders [23,24,25].

Although the mechanism of brain injury and the emergence of CSD following SAH are generally understood, much more study is still required to clarify the fundamental questions surrounding the onset and progression of CSD. Moreover, developing a sensible and efficient clinical intervention plan is challenging given the complexity of its mechanism. Current research has identified a number of potential intervention targets, such as increasing the availability of NO to raise the CSD threshold [26], Mg ion antagonism [27], NMDA receptor antagonists [28], and high perfusion pressure to shorten the duration of CSD [29]. However, in practice, multi-directional intervention may be required to improve prognosis.

### 1.3. Immune Inflammation

Brain injury and CVS after SAH are both significantly influenced by the immune–inflammatory response. Despite the presence of certain immune mechanisms in the central nervous system, the disruption of the blood–brain barrier (BBB) allows for the activation of inflammatory cells, upregulation of immune molecules, and release of inflammatory mediators in response to the stimulation of blood lysate antigens. Following SAH, lymphocytes and macrophages invade the blood vessels and brain tissue. VCAM-1, ICAM-1, and selectin are chemokines that play a role in the injury to the brain parenchyma and microvascular system by mediating the adherence of lymphocytes and platelets to endothelial cells [30]. By preventing lymphocyte migration, these chemokines can reduce CVS and immune-related brain injury. Neutrophils infiltrate cerebral blood arteries and brain parenchyma as well. By preventing neutrophil activation, early microvascular and brain parenchyma damage in SAH can be substantially reduced [30,31].

Infiltrating inflammatory cells and activated microglia in the central nervous system produce a multitude of inflammatory cytokines. Studies show that IL-1, IL-6, and TNF-α levels are significantly elevated during the acute stage of SAH, and medication treatment targeting TNF-α, IL-1, and their receptors can significantly enhance EBI. Ras-MAPK-NF-κB, JAK/STAT, TLR4/NF-κB, and other signaling pathways are all engaged in the process of transmitting immunological signals to the production of inflammatory factors. To gain a deeper understanding of the mechanism underlying the immune–inflammatory cascade reaction and develop a targeted intervention strategy, one of the current research areas involves interfering with and knocking off the key nodes of each signaling pathway [32,33].

Damage-associated molecular patterns (DAMPs) are accountable for the inflammatory response that occurs following SAH, can induce the discharge of high-mobility group box 1 (HMGB1), and can activate downstream TLR/NF-κB [34,35,36]. In the initial phase after SAH, HMGB1 may be briefly and passively liberated by necrotic and damaged cells, while in the later stages, inflammatory mediators can stimulate inflammatory and nerve cells to actively release HMGB1, which can exacerbate brain injury. Therefore, HMGB1 may play a crucial role in DAMPs after SAH, may take a significant part in immunological brain injury, and is closely linked to the prognosis as well as the activation of immune inflammation that results from antigen exposure.

According to recent research, immunological inflammation’s main contribution to neuronal dysfunction is BBB degradation, which exacerbates cerebral edema and encourages mitochondrial pathway apoptosis. Inflammatory mediators have the ability to activate the cell necrosis pathway as well as the apoptosis pathway. TNF-α, for instance, interacts with receptor-associated factor, receptor-interacting protein 1, and Fas-associated death domain protein (FADD) to activate its receptor, which subsequently activates caspase-8 to trigger an apoptosis-cascade reaction. Additionally, RIP1, RIP3, and FADP work together to initiate the necroptosis pathway when caspase-8 activity is suppressed.

### 1.4. Cell Death

The death of neurons and endothelial cells is the primary histocytological and molecular biological indicators in the study of EBI following SAH, and it offers significant guidance for assessing and enhancing neurological prognosis. Because apoptosis can be observed minutes after SAH and can persist for up to 24 h [37], cerebral ischemia brought on by increased ICP may be the initial element to cause the pathway to be activated. The mechanism of apoptosis after SAH has been gradually linked to various brain injury mechanisms, including microvascular changes after SAH, cellular energy metabolism stress, CSD, inflammatory injury, and excitotoxicity, based on years of research progress.

The two types of apoptotic pathways are extrinsic (death receptor pathways) and intrinsic (capacity-dependent and mitochondrial pathways). The intrinsic pathway is supported by a variety of apoptosis-related proteins in mitochondria, including cytochrome C, apoptosis-inducing factor (AIF), second mitochondrial-derived caspase activator/apoptosis inhibitor protein direct binding protein, nucleic acid Dicer G, and the activation of various caspase precursor proteins; particularly significant indicators of the mitochondrial route are the cascade response of the caspase family and the final activation of caspase-3 [38]. A range of cell membrane receptors, including TNFR, Fas, and DR3–5, as well as its ligands TNF-α, TRAIL, and Fas, activate the death receptor pathway. To activate the caspase cascade reactions and control cell apoptosis, death receptors join forces with FADP and caspase-8 to form death-signaling complexes. Numerous studies show that TNF-α and its receptors are upregulated following SAH, and blocking this pathway has positive neuroprotective benefits [39]. Neuronal or endothelial cell apoptosis is regulated by non-caspase-dependent endogenous mechanisms, such as PARP/AIF, BNIP3, and the endoplasmic reticulum pathway [40].

When cells are under stress or hunger, a process known as autophagy transforms inactive macromolecules, damaged or redundant organelles, and other materials into active tiny molecules by producing autophagosomes and fusing with lysosomes to generate autophagy lysosomes. Although excessive autophagy is necessary to meet energy requirements and maintain the integrity of the intracellular environment, it can also result in autophagic cell death. In the investigation of cerebral ischemia, it was discovered that inhibiting autophagy leads to two radically different outcomes, encouraging neuronal survival or death, which may be related to the degree and length of injury. The ultrastructure of neurons and astrocytes was examined using electron microscopy, and autophagy-related proteins were found in cortical brain tissue. After SAH, it was discovered that autophagy pathways were active in EBI [41]. According to a different study [42], drug-activated autophagy might lessen brain injury-related markers, whereas autophagy inhibition can worsen brain damage. This suggests that autophagy may be neuroprotective in SAH due to its ability to prevent apoptosis.

Necroptosis, which is characterized by autophagy activation and necrotic cell shape, occurs when the apoptotic pathway is suppressed. The modulation of this signaling pathway may hold promising therapeutic implications given the demonstrated involvement of necroptosis in cerebral ischemia and traumatic injury in recent investigations [43]. Necroptosis is mediated by death receptors and RIP with the interaction of RIP3 and RIP1 as the key mechanism. After the death receptor is activated, RIP1 can mediate death receptor-related apoptosis together with FADP and caspase-8, but when the caspase pathway is inhibited, for example, when the broad-spectrum caspase inhibitor zVAD is applied, it will form a RIP1–RIP3 combination with RIP3. Due to the complexity of the process, there is an increase in energy metabolism and the production of reactive oxygen species (ROS), which damage organelles and the membranes of cells and lead to programmed necrosis. After a SAH brain injury, numerous signaling pathways are engaged in mediating cell death, and the cell death that is observed at any point is the product of numerous causes and numerous pathways. For instance, 24 h after SAH, necrosis and apoptosis can be observed simultaneously with apoptosis, deep cortical apoptosis, and superficial cortical autophagy activation [41]. The degree of the injury and the cell’s internal and exterior surroundings eventually determine the cell death pathway. Even though necrosis may still predominate in the majority of cells, and there may be variations in the main pathways in different regions and at different times, targeted manipulation of certain pathways can improve neuroprotection.

### 1.5. Damage to the BBB and Cerebral Edema

Multiple pathophysiological processes are involved in BBB disruption following SAH, among which endothelial cell apoptosis is implicated [44]. BBB disruption can be facilitated by blood breakdown products such as oxyhemoglobin and the associated oxidative stress. Inflammatory cytokines, such as TNF-α, IL-1β, and thromboxane A2, can also compromise the BBB by encouraging endothelial cell death and activating matrix metalloproteinases.

Matrix metalloproteinase-9 (MMP-9) is implicated in early BBB degradation after SAH according to a growing body of research [45]. MMP-9 is responsible for the degradation of the extracellular matrix of the basement membrane of brain microvessels, such as type IV collagen, laminin, fibronectin, as well as tight junction proteins such as ZO-1 and claudin-5 located between endothelial cells [46]. Basement membrane degradation first began 6 h after the rat intravascular-puncture model, and reached the highest after 48 h [47] accompanied by the up-regulation of MMP-9. After the BBB is destroyed, the fluid/protein extravasates into the interstitium, and blood lysates and inflammatory factors participate in brain damage through the BBB, resulting in the development of cerebral edema.

Cerebral edema is a frequent pathological manifestation of SAH in both animal models and human patients, and global cerebral edema is identified as a notable independent risk factor for unfavorable outcomes. Studies show that, in addition to BBB dysfunction, brain edema after SAH is also closely related to aquaporin 4 (AQP4). Drug inhibition of AQP4 expression can reduce cerebral edema, but gene knockout of AQP4 aggravates cerebral edema [48], suggesting the importance of AQP4 in maintaining brain tissue water balance after SAH and the value of further research.

### 1.6. Oxidative Stress

The release of hemoglobin cleavage products into the subarachnoid space following SAH triggers oxidative stress, which is further exacerbated by cerebral ischemia–reperfusion in the acute stage of the condition. Oxidative stress promotes lipid peroxidation and protein oxidation, which cause DNA damage, activate apoptosis pathways, and trigger inflammatory cascade reactions. Increased severity of brain injury is associated with its involvement in numerous pathological processes, including apoptosis, blood–brain barrier (BBB) disruption, vascular smooth muscle injury, and endothelial injury. These processes collectively contribute to the progression and exacerbation of brain injury [49]. Reducing oxidative stress by preventing apoptosis and BBB degradation can lower EBI [50].

Nitric oxide synthase (NOS), nicotinamide adenine dinucleotide phosphate (NADPH) oxidase, and heme oxygenase are all involved in the generation of reactive oxygen species (ROS) following SAH. In the context of oxidative stress, NO produced by NOS, particularly endothelial NOS (eNOS), interacts with superoxide anions to form excess peroxynitrite, which further causes eNOS to decouple and generate more superoxide anions. After the first bleed, cerebral NO dramatically dropped, which may be what primarily causes acute CVS, but it significantly increased after 24 h. NO metabolites also showed comparable expression patterns [51], which demonstrates that NOS and NO are crucial in oxidative stress-related brain damage. Recent research shows that oxidative stress damage can be efficiently reduced by removing the triggers for ROS creation, blocking ROS generation, and increasing antioxidant reactions [52,53]; however, more investigation is necessary to evaluate the practical relevance of this discovery.

### 1.7. Nitric Oxide Synthase

Blood artery dilation, anti-platelet activation, and inflammation suppression are all effects of NO. Following SAH, NO showed a drop in the acute phase, took part in acute CVS, rose, and was crucial for EBI. The uncoupling of eNOS, which is linked to an abnormal rise in NO expression and a decrease in NO availability as a result of eNOS failure, is another important mechanism of NO involved in CVS and brain injury. It is characterized by an increase in nitrification stress and oxidative stress [54]. In the study of cerebral ischemia, it was found that knockout of neural NOS (nNOS) and inducible NOS (iNOS) could improve cerebral infarction, and knockout of eNOS increased infarct size [55]. In SAH, eNOS knockout can reduce microvascular damage, inhibit neurodegeneration, and result in other protective effects [56], which is opposite to the role of eNOS in improving large vessel spasm and cerebral ischemic infarction [57]. The neurovascular effects of NO raise new questions.

### 1.8. Role of Microglial Cells and Immune Cells in EBI

Microglial cells and immune cells play crucial roles in the pathophysiology of EBI. These cells are involved in neuroinflammatory processes and the release of various inflammatory mediators, which can exacerbate brain damage following subarachnoid hemorrhage (SAH). Activation of microglia and the infiltration of immune cells have been observed in experimental models and clinical studies of SAH [58]. Furthermore, microglial activation and immune cell infiltration are associated with the release of pro-inflammatory cytokines, reactive oxygen species, and other neurotoxic factors, leading to neuronal cell death and tissue damage [59]. These processes can contribute to the progression of secondary brain injury following SAH.

## 2. Developments in the Study of Brain Injury Following SAH

In animal research, targeted therapies that addressed the aforementioned brain damage processes produced a variety of positive outcomes, some of which were also corroborated by clinical investigations. Despite the fact that DCVS and DCI continue to be the subject of clinical studies, diagnoses, and treatments, it is challenging for the research and intervention strategies developed for the two to produce adequate outcomes. Only nimodipine is commonly prescribed to treat SAH, which is a calcium channel blocker. It primarily affects the smooth muscle cells of the brain’s blood vessels, helping to relax and widen them to improve blood flow and reduce the risk of complications from SAH and is acknowledged to be able to greatly improve clinical patients’ prognoses after examining many years of research; this ability is independent of its easing influence on CVS [60]. Therefore, it is imperative to develop an innovative treatment strategy for SAH that focuses on the mechanisms associated with early brain injury (EBI). During the initial stages following the stabilization of patients with SAH, a comprehensive approach involving vasodilation, optimization of cerebral blood flow (CBF) and metabolism, enhancement of the anti-inflammatory response, reduction of oxidative stress, inhibition of platelet aggregation, and other relevant interventions is crucial. This multifaceted approach aims to effectively inhibit multiple critical brain injury mechanisms, thereby providing enhanced and substantial neurovascular protection and ultimately influencing patient prognosis. The prevention of EBI following SAH and the treatment of delayed complications have both benefited from numerous recent breakthroughs in animal studies and clinical trials.

### 2.1. Calcium Channel Antagonists and Endothelin-1 Receptor Blockers

The FDA has given the most popular calcium channel blocker for SAH, nimodipine, permission to be used in clinical settings. According to studies conducted on animals, nimodipine can effectively reduce artery constriction and boost cerebral blood flow when applied within six hours following a SAH. Nimodipine is shown to lower the likelihood of ischemia consequences and their poor prognosis in clinical patients. Nimodipine’s involvement in reversing CVS does not appear to be implicated despite the fact that the mechanism by which it enhances patient outcomes is not well understood. Nimodipine-treated patients demonstrated little angiographic reduction in vasospasm [61]. Therefore, more research on nimodipine’s impact on the neurovascular unit’s functionality is required to understand how it protects the brain.

Because ET-1 promotes CVS following SAH, reducing ET-1 levels, inhibiting ET-1 receptors, and limiting endothelin receptor activation can all be effective approaches to lessen CVS. ET-1 receptor blockers are now thought to be the most effective strategy. Early use of ET-1 receptor antagonists following SAH was shown to successfully restore CBF in animal tests. Clinical investigations demonstrated that the ET receptor antagonist clozosentan can reverse CVS without significantly changing the prognosis of the patient [5]. The observed effect could be attributed to the drug’s toxicological impacts, or it could be related to the crucial role of early brain injury (EBI) in the patient’s survival and neurological recovery. In this context, the early intervention treatment approach aims to optimize neurological recovery by addressing EBI’s underlying mechanisms.

### 2.2. Antiplatelet Drugs

The most popular antiplatelet medications studied in SAH are acetylsalicylic acid (aspirin) and ticlopidine. Meta-analysis revealed that patients treated with antiplatelet medications tended to have a better prognosis than patients who did not receive antiplatelet medications [62]. To ascertain the dose–time effect of antiplatelet drugs like aspirin on the prognosis of SAH, larger trials are necessary.

### 2.3. Anti-Inflammatory Drug

Drugs that can be applied to humans and animals in SAH anti-inflammation include non-steroidal anti-inflammatory drugs, immunosuppressants, glucocorticoids, serine protease inhibitors, and antioxidant drugs. Animal studies have investigated a variety of therapeutic approaches for treating various elements of immune inflammation, including blocking the activation of immune cells [63], opposing inflammatory substances [32,64], focusing on inflammatory signaling pathways [65], and others. Although they have not been used in clinical settings, they established a theoretical framework for efficient screening of particular targets for immune brain damage.

### 2.4. Statins

Statins can target and affect various aspects of the brain injury mechanism after SAH and play a neuroprotective role. Statin therapy can lessen both early and delayed brain damage in the acute period following SAH (within minutes of onset in animals and within three days in humans). For example, statins are known to exhibit various beneficial effects in the context of SAH. They can inhibit cholesterol synthesis, impede platelet aggregation, suppress glutamate excitotoxicity, alleviate apoptosis of endothelial cells and neurons, reduce inflammation, promote angiogenesis, increase the expression and activation of endothelial nitric oxide synthase (eNOS), and improve vascular function among other mechanisms [66,67]. However, there is ongoing debate over whether statin medication for individuals with acute SAH is beneficial, and meta-analysis cannot produce consistent results. It is still unclear whether statin use can enhance neurological outcomes in patients with SAH. Conclusions might be made based on a bigger randomized controlled trial that takes into account statins’ dual effects on EBI and delayed consequences [68].

### 2.5. Antioxidant Stress Drugs

In animal studies, a number of antioxidant stress medications, including iron chelators, lipid peroxidation inhibitors, peroxidase, and other free radical scavengers, demonstrated effective protective benefits [69]. Methylprednisolone and tirazart are commonly used medications in clinical settings due to their known antioxidant effects. However, it is important to note that their efficacy cannot be solely assessed based on clinical research alone [70,71]. Studies on the free radical scavenger edaravone were conducted in CVS but not as many as in EBI. According to animal studies, edaravone usage within 24 h can lower lipid peroxidation, inhibit caspase-3, lower mortality, and enhance neurological prognosis. While no significant difference was observed in the risk of delayed ischemic neurological dysfunction (DIND) in clinical trials, edaravone was found to reduce the incidence of cardiovascular events, cerebral infarction, and delayed ischemic brain injury four days after subarachnoid hemorrhage (SAH). These findings suggest that earlier clinical intervention may help elucidate the therapeutic potential of edaravone in this context [72,73].

## 3. Main Problems and Prospects

The pathogenesis of EBI involves a number of pathological processes, including acute ICP increase, CPP decrease, microcirculation change, ion imbalance, CBF decrease, brain metabolic crisis, oxidative stress, immune inflammation, and BBB injury among others, and these processes interact with one another, promote nerve cell death and cerebral edema, and also have a significant impact on the progression of the disease. There is still a dearth of thorough, systematic, and critical research results on the mechanism of the occurrence and progression of EBI after SAH, and there are still many important issues to be resolved, even though reasonable intervention strategies for these early pathological processes have achieved some beneficial effects in basic research and clinical trials. In order to fully understand the molecular basis and the ideal outcome of intervention therapy, it is imperative to conduct extensive basic and clinical research, change one’s way of thinking, organically combine macro and micro, and develop across a wide range of disciplines.

The conventional cardiovascular (CVS) mechanism is no longer the primary focus of current SAH research. Instead, there is a shift towards investigating neurological function. Extensive studies have elucidated various intricate pathophysiological mechanisms associated with SAH, such as cerebral ischemia and hypoxic injury following brain injury, toxicity resulting from the breakdown of blood products, activation of neuroimmune inflammation, acute microvascular changes, cortical spreading depression (CSD), activation of different cellular signaling pathways, and more, which interact to promote the occurrence and development of the disease. In order to lessen brain damage after SAH and encourage brain repair, it is important to focus on the key issues in brain injury and brain protection after SAH, continuously reveal the evolution rules of these pathological processes, integrate neuroprotective strategies in various directions, and speed up drug pharmacology and toxicology research. Brighter possibilities result from raising patient quality of life and survival rates.

## Data Availability

The data sets analyzed during the current study are available from the corresponding author on reasonable request.

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
