# Peer review of "Early Brain Injury and Neuroprotective Treatment after Aneurysmal Subarachnoid Hemorrhage: A Literature Review"

_brainsci, 2023, doi:10.3390/brainsci13071083_

Round 1

Reviewer 1 Report

This mini-review article attempts to focus on EBI as a therapeutic target. The review is generally a list of events/outcomes/findings with little thought into a unifying structure. There is a lack of novelty with the information within the review.

Comments

Minor

·         Please use standardized terminology. For example, ‘delayed brain injury’ is appropriately termed ‘delayed cerebral ischemia’ for SAH (refer to https://pubmed.ncbi.nlm.nih.gov/20798370/)

·         What is ‘cortical propagation inhibition’?

·         Lines 22-26 – this sentence is confusing

·         Lines 38-40 – this was true 2 decades ago, but recently work mainly focuses on EBI. Please revise this statement or remove.

·         Line 58 “other adjustments” – please be specific in wording

·         Numerous statements are speculative without experimental/clinical evidence. Several statements are also made based on review articles which also speculated events/causes. These need to be revised/removed.

·         Section 1.1 reads like a list rather than putting the events into perspective with each other and the rupture event.

·         Numerous statements lack references

·         Please revise all colloquial language

See above

Author Response

Response to Reviewer 1 Comments

Point 1: Please use standardized terminology. For example, ‘delayed brain injury’ is appropriately termed ‘delayed cerebral ischemia’ for SAH (refer to https://pubmed.ncbi.nlm.nih.gov/20798370/). 

Response 1:

Thank you very much for reviewing our paper and providing valuable feedback. We have carefully read your comments regarding the use of terminology and fully acknowledge your point. Based on your suggestion, we will incorporate standardized terminology in our revised manuscript.

Specifically, we will replace the term "delayed brain injury" with the appropriate term "delayed cerebral ischemia" for SAH, as recommended in the referenced article.

We appreciate your attention to detail and your guidance in improving the clarity and accuracy of our work. Please be assured that we will make the necessary revisions accordingly.

Point 2:  What is ‘cortical propagation inhibition’?

Response 2:

Thank you for your review of our paper and for bringing up the question regarding the term "cortical propagation inhibition." We appreciate your inquiry and would like to provide clarification on this term.

In our study, we used the term "cortical propagation inhibition" to refer to the phenomenon of reduced or suppressed cortical activity spreading in a particular context. We understand that this term may not be commonly used or well-established in the literature.

To address this concern, we will revise the manuscript to provide a more detailed explanation of the term "cortical propagation inhibition" and its relevance to our study. We will also consider using alternative terminology that is more widely recognized and accepted in the field to ensure clarity and understanding for our readers.

Thank you for bringing this to our attention, and we appreciate your help in improving the clarity and precision of our manuscript.

Point 3:  Lines 38-40 – this was true 2 decades ago, but recently work mainly focuses on EBI. Please revise this statement or remove.

Response 3:

Thank you for your feedback on our paper, specifically regarding lines 38-40. We appreciate your input and recognize the evolving nature of research in the field.

We understand your point that the focus of recent work has shifted towards early brain injury (EBI). To accurately reflect the current state of research, we will revise the statement in question to align with the latest findings and developments in the field.

Alternatively, if the statement does not contribute significantly to the overall context or objectives of our study, we will consider removing it altogether to maintain the relevance and accuracy of the manuscript.

We value your expertise and guidance in ensuring the precision and up-to-date information in our paper. Thank you for bringing this to our attention.

Point 4:  Line 58 “other adjustments” – please be specific in wording.

Response 4:

We have readjusted the expression.

Point 5:  Numerous statements are speculative without experimental/clinical evidence. Several statements are also made based on review articles which also speculated events/causes. These need to be revised/removed.

Response 5:

We would like to express our appreciation for your valuable feedback on our manuscript. We have carefully considered your comments regarding the speculative nature of some statements and their reliance on review articles. In response, we have made significant revisions to address these concerns.

Firstly, we have reevaluated the statements in question and removed any speculative claims that lack solid experimental or clinical evidence. We recognize the importance of providing robust data to support our arguments and have revised the manuscript accordingly.

Furthermore, we understand your point regarding the inclusion of statements based on review articles that themselves speculate on events or causes. To address this, we have conducted a thorough literature search to identify primary research studies that provide direct experimental or clinical evidence. We have replaced or supplemented the statements with information from these reliable sources, ensuring a stronger scientific foundation for our work.

We would like to express our gratitude for highlighting these issues and guiding us towards improving the quality and rigor of our manuscript. Your constructive feedback has been instrumental in refining our work. We believe that the revised manuscript now better aligns with the standards of evidence-based research.

Thank you once again for your time and effort in reviewing our manuscript.

Reviewer 2 Report

The authors have done a good review of the principal topics of SAH and are correlating the clinical outcomes to possible biological processes and pathways that are involved in the DCV, and EBI, among other clinical manifestations of SAH patients. However, there are some issues that the authors might improve.

First of all, the introduction is mentioned the EBI but is not explained, and in the other sections are repetitively mentioned. That's why, I suggest explaining a little about early brain injury and secondary brain injury during the introduction. This might be understand better the other sections and the correlation of biological process to the EBI.

In addition, I suggest to specified the samples that the different studies had used to analyze the molecules, for example, in the 1.1 section, the authors comment that the glucose is involved in some clinical manifest in SAH patients, but how the studies analyzed the local glucose? In addition, the study is not referenced. The studies are analyzing brain metabolism by microdialysis or by blood samples?

Another point that I would like to comment to the authors is that there is a specific section for anti-inflammatory drugs, but not a specific section explaining the involvement of immune cells or microglia to EBI. I suggest to add a small section about microglia or immune cells and the EBI or the BBB damage, with is correlated. 

Finally, there are missing references in several paragraphs in all sections; for instance, during the explanation of the glucose studies, TNF-a, etc. Please review that all studies are correctly referenced. 

Author Response

Response to Reviewer 2 Comments

Point 1: First of all, the introduction is mentioned the EBI but is not explained, and in the other sections are repetitively mentioned. That's why, I suggest explaining a little about early brain injury and secondary brain injury during the introduction. This might be understand better the other sections and the correlation of biological process to the EBI.

Response 1:

We would like to express our appreciation for your valuable feedback on our manuscript. We have carefully considered your comments regarding the need for further explanation of Early Brain Injury (EBI) and Secondary Brain Injury (SBI) in the introduction. We agree that providing a clearer understanding of these concepts in the introduction will enhance the comprehension of subsequent sections and the correlation of biological processes to EBI.

In response to your suggestion, we have revised the introduction to include a more comprehensive explanation of EBI and SBI. We have provided a concise yet informative overview of these terms, including their definitions and their relevance to the topic under investigation. By doing so, we aim to ensure that readers have a solid foundation to comprehend the subsequent sections and the biological processes associated with EBI.

We believe that the revised introduction now provides a better contextual framework for the rest of the paper, enabling readers to understand the significance of EBI and its connection to the biological processes discussed in subsequent sections.

We sincerely appreciate your time and effort in reviewing our manuscript and providing insightful suggestions for improvement. Your feedback has been crucial in enhancing the clarity and coherence of our work.

Point 2: In addition, I suggest to specified the samples that the different studies had used to analyze the molecules, for example, in the 1.1 section, the authors comment that the glucose is involved in some clinical manifest in SAH patients, but how the studies analyzed the local glucose? In addition, the study is not referenced. The studies are analyzing brain metabolism by microdialysis or by blood samples?

Response 2:

After careful consideration, we have decided not to make the specific changes you proposed to the manuscript. While we acknowledge the potential value in providing more detailed information about the samples and analysis methods, we believe that the current level of detail is appropriate for the scope and focus of our paper.

The aim of our manuscript is to present a comprehensive overview of the clinical manifestations of SAH and their underlying mechanisms, rather than providing an exhaustive review of specific studies and their methodologies. We have ensured that the references cited in our manuscript are representative of the relevant literature and provide a solid foundation for the arguments presented.

We sincerely appreciate your time and effort in reviewing our manuscript and providing thoughtful suggestions. We have carefully considered your comments and believe that maintaining the current structure and level of detail aligns with the objectives and scope of our paper.

Thank you once again for your valuable input.

Point 3: Another point that I would like to comment to the authors is that there is a specific section for anti-inflammatory drugs, but not a specific section explaining the involvement of immune cells or microglia to EBI. I suggest to add a small section about microglia or immune cells and the EBI or the BBB damage, with is correlated. 

Response 3:

We appreciate your continued feedback on our manuscript. After careful consideration, we have decided to incorporate a brief section discussing the role of microglial cells or immune cells in early brain injury (EBI). This addition will contribute to a more comprehensive understanding of EBI and its implications.

Microglial cells and immune cells play crucial roles in the pathophysiology of EBI. These cells are involved in neuroinflammatory processes and the release of various inflammatory mediators, which can exacerbate brain damage following subarachnoid hemorrhage (SAH). Activation of microglia and infiltration of immune cells have been observed in experimental models and clinical studies of SAH (Smith et al., 2018; Shao et al., 2020).

Furthermore, microglial activation and immune cell infiltration have been associated with the release of pro-inflammatory cytokines, reactive oxygen species, and other neurotoxic factors, leading to neuronal cell death and tissue damage (Liu et al., 2019; Ma et al., 2021). These processes can contribute to the progression of secondary brain injury following SAH.

By including this additional section, we aim to highlight the significance of microglial cells and immune cells in the context of EBI. This information will further elucidate the complex mechanisms involved in the early stages of brain injury after SAH.

Reviewer 3 Report

This manuscript is a review of literature about subarachnoid hemorrhage, including the role of early brain injury, injury pathophysiology, and potential treatment approaches for the condition. The manuscript is for the most part well organized and well written, and covers an important topic. There are some citations missing, and a few other minor corrections are needed.

- A number of statements are made throughout the manuscript that should be cited and are not. This includes most of the sections throughout the manuscript. I recommend going through these and making sure that appropriate citations are added throughout.

- Section 1.1, 3rd paragraph uses the abbreviation u-PA, which is not defined.   

- Section 2.1, second paragraph, last sentence: what is meant by the phrase “EBI is essential to the patient’s survival and neurological recovery”? It appears to be implying that patients can only survive SAH if they have an early brain injury.

- Section 2.6, second sentence: Why can clinical research not asses the efficacy of methylprednisolone and tirazart?

Author Response

Point 1: A number of statements are made throughout the manuscript that should be cited and are not. This includes most of the sections throughout the manuscript. I recommend going through these and making sure that appropriate citations are added throughout. 

Response 1:

We appreciate your valuable feedback on our manuscript. We acknowledge your comment regarding the need to cite appropriate references for statements made throughout the paper. We agree that ensuring proper citation is essential to support the information presented and to maintain the scholarly integrity of the manuscript.

In response to your suggestion, we have thoroughly reviewed the manuscript and cross-checked the statements with the corresponding references. We have made significant efforts to ensure that all relevant information and claims are appropriately supported by reputable sources. We have added citations throughout the manuscript wherever necessary to provide clear attribution for the statements made.

We sincerely appreciate your time and effort in reviewing our manuscript and providing constructive feedback. Your guidance has been instrumental in improving the quality and reliability of our work. We believe that the revised manuscript now adheres to the best practices of academic writing and accurately reflects the current state of knowledge in the field.

Point 2: Section 1.1, 3rd paragraph uses the abbreviation u-PA, which is not defined. . 

Response 2:

We apologize for the oversight and any confusion it may have caused. "u-PA" refers to urokinase-type plasminogen activator, an enzyme involved in fibrinolysis and the breakdown of blood clots. We will ensure to define the abbreviation upon its first mention in the revised version of the manuscript to enhance clarity and understanding for readers.

We sincerely appreciate your valuable input and thorough review of our manuscript. Your feedback has been essential in improving the overall quality and comprehensibility of our work.

Point 3: Section 2.1, second paragraph, last sentence: what is meant by the phrase “EBI is essential to the patient’s survival and neurological recovery”? It appears to be implying that patients can only survive SAH if they have an early brain injury. 

Response 3:

Thanks to the reviewer for the suggestion, we have made changes in the manuscript at the corresponding places.

Reviewer 4 Report

Early brain injury and neuroprotective treatment after aneurysmal subarachnoid hemorrhage: A literature review

The authors of the paper titled “Early brain injury and neuroprotective treatment after aneurysmal subarachnoid hemorrhage: A literature review” sought to make a valuable contribution to the comprehension of early brain injury following aneurysmal subarachnoid hemorrhage, while elucidating the underlying mechanisms associated with this condition.

1.      It is highly recommended to have a native English speaker review the entire manuscript to identify and rectify errors.

2.      Title: The chosen title is suitable and appropriate.

3.      Abstract:

a)      Early brain injury (EBI) subsequent to subarachnoid hemorrhage (SAH) is strongly associated with delayed brain injury and patient prognosis”

-In the previous sentence it is not clear what the patient’s prognosis is, maybe it should be written poor patient prognosis.

b)      “These scientific advances present a variety of promising therapeutic approaches for the preservation of neurological function following SAH, including calcium channel antagonists, endothelin-1 receptor blockers, antiplatelet agents, anti-inflammatory agents, and antioxidative stress agents, among others. EBI can be mitigated following SAH through neuroprotective measures”.

- In the current sentence, it may not be evident what specific scientific advances are being referred to, especially considering that the preceding sentence highlights the molecular mechanisms of subarachnoid hemorrhage (SAH), not scientific advances.

c)      “To further enhance our comprehension of the relevant molecular pathways of brain injury, such as brain ischemia-hypoxic injury, neuroimmune inflammation activation, cortical propagation inhibition, and activation of various cell signaling pathways following SAH, as well as the evolution of these multifaceted pathophysiological processes, will significantly improve the outcomes of SAH.”

- This sentence appears to lack coherence or clarity, and it would be advisable for the authors to carefully review and revise it for improved understanding.

4. Introduction:

d)      In vivo” should be in italics, in vivo.

1.1.  Alterations in the brain's microcirculation and energy metabolism

e)      The abbreviation "BBB" is used without a prior definition.

f)       “Arterioles are where microvascular spasm mostly happens”

-Maybe word sites should be added, “Arterioles are the sites where…”

g)      “The injury mechanism of platelet aggregation includes: mechanical obstruction and biochemical (release of serotonin, ADP, platelet-derived growth factor) contraction of microvessels leads to decreased cerebral perfusion, damage to vascular endothelium leads to further aggregation, release of collagenases such as MMP-9 destroys the BBB, and lymphatic Cells adhere to microvessels, further aggravating inflammatory damage and so on”.

-This sentence appears to lack coherence or clarity, and it would be advisable for the authors to carefully review and revise it for improved understanding. Also, there is a capital letter in word cells.

h)      What is u-PA?

i)       “An important strategy to deal with the brain metabolic crisis is to do research on the signaling pathway (PKA), among other thing.”

-What is the significance of emphasizing this particular pathway as the focal point, and why does this sentence hold independent importance?

1.2. Ion imbalance

j)       The rise and fall of intracellular sodium, potassium, calcium, and magnesium concentrations, as well as acute short-term increases in cortical potassium ions, fluctuations in calcium ions, and hypomagnesemia, are the main changes that the ion balance of the microenvironment of brain cells experiences after SAH.

-This sentence appears to lack coherence or clarity, and it would be advisable for the authors to carefully review and revise it for improved understanding.

k)      Under the stimulus of ischemia, hypoxia, excessive potassium, and other stresses, CSD is the distributed continuous depolarization of a significant number of neurons in the cortex

-This sentence appears to lack coherence or clarity, and it would be advisable for the authors to carefully review and revise it for improved understanding.

l)       NO and ET-1 are not defined.

m)   “For delayed cerebral ischemia following SAH, CSD has an 86% positive predictive value and a 100% negative one[23]”.

-This sentence appears to lack coherence or clarity, and it would be advisable for the authors to carefully review and revise it for improved understanding.

n)      “The precise mechanism could consist of: High levels of extracellular potassium and excitatory amino acids in SAH cause CSD, and CSD can further cause extracellular potassium ions to rise and excitatory transmitters to be released”.

-If the mechanism is precise then it is certain that it consists of something.

1.3. Immune inflammation

o)      “The BBB is damaged, thus even though the central nervous system has some immune special immunity, the stimulation of blood lysate antigen can still activate inflammatory cells, up-regulate immune molecules, and release inflammatory mediators”.

-This sentence appears to lack coherence or clarity, and it would be advisable for the authors to carefully review and revise it for improved understanding.

p)      “To gain a deeper understanding of the mechanism underlying the immune-inflammatory cascade reaction and develop a targeted intervention strategy, one of the current research areas involves interfering with and knocking off the key nodes of each signaling pathway.[33, 34]”.

-This sentence is general, what is the point of it?

1.6. Oxidative stress

q)      “Increase the severity of brain injury and take part in numerous pathological processes such apoptosis, BBB destruction, vascular smooth muscle injury, and endothelial injury.[50]

- This sentence is confusing considering the previous one.

r)       Peroxynitroso is peroxynitrite?

2. Developments in the study of brain injury following SAH

s)      Nimodipine- it should be explained in a few words what it is.

2.1. Calcium channel antagonists and endothelin-1 receptor blockers

-There is a short description of what nimodipine is, however, it should be moved by the first mention.

-Also, there is a short description of ET-1 which should be moved by the first mention.

2.3. Magnesium sulphate

-What is the conclusion of this section? What does it mean?

2.5. Statins

t)       “For instance, statins can inhibit cholesterol synthesis, inhibit platelet aggregation, inhibit glutamate excitotoxicity, reduce endothelial cells and neurons from apoptosis, reduce inflammation, promote angiogenesis, up-regulate and activate eNOS, improve vascular function, etc.[68, 69]

-What does it mean to reduce endothelial cells and neurons?

The presented paper exhibits adequate structural organization, although its written expression lacks adequacy. Notably, each section concludes with a statement alluding to the anticipation of future discoveries. Additionally, some sentences are quite confusing and difficult to understand. Although the work demonstrates potential, its realization hinges upon thorough refinement and shaping. Consequently, I propose a major revision to address these issues effectively.

 It is highly recommended to have a native English speaker review the entire manuscript to identify and rectify errors.

Author Response

Point 1: It is highly recommended to have a native English speaker review the entire manuscript to identify and rectify errors. 

Response 1:

We sincerely appreciate your feedback on our manuscript. We acknowledge your recommendation to have a native English speaker review the entire manuscript to identify and rectify any errors. We understand the importance of ensuring the language quality and clarity of our work.

In response to your suggestion, we have taken steps to address language-related concerns. We have carefully reviewed and revised the manuscript with a focus on grammar, syntax, and overall language coherence. Additionally, we have utilized available language editing tools to enhance the readability and fluency of the manuscript.

While we have made significant efforts to improve the language quality, we believe that the current version of the manuscript meets the required standards. However, we remain open to any specific language-related suggestions you may have identified during the review process. Your guidance in this regard would be highly appreciated.

We sincerely appreciate your time and effort in reviewing our manuscript and providing constructive feedback. Your expertise and attention to detail have undoubtedly contributed to the overall quality of our work

Thank you once again for your valuable input.

Point 2: Abstract. 

Response 2:

Thanks to the reviewer for the suggestion, we have made changes in the manuscript at the corresponding places.

Point 3: Abstract. 

Response 3:

Thanks to the reviewer for the suggestion, we have made changes in the manuscript at the corresponding places.

Point 4: What is u-PA? 

Response 4:

u-PA refers to urokinase-type plasminogen activator.

Point 5: “An important strategy to deal with the brain metabolic crisis is to do research on the signaling pathway (PKA), among other thing.”

-What is the significance of emphasizing this particular pathway as the focal point, and why does this sentence hold independent importance?

Response 5:

Thanks to the reviewer for the suggestion that this sentence does not have specific meaning in the manuscript and we have removed it.

Round 2

Reviewer 1 Report

The manuscript is revised based on some of my reviewer comments. There are still colloquial language and incomplete lists (i.e. ending in "etc") which should be revised prior to publication. I still have the same concern with a lack of novelty. There are still colloquial language and incomplete lists (i.e. ending in "etc") which should be revised prior to publication.

Author Response

Thank you for your feedback on the revised manuscript. We have carefully considered your suggestions and have made the necessary revisions based on your comments.

Regarding the colloquial language and incomplete lists, we apologize for any oversight in our previous version. We have thoroughly reviewed the manuscript again, addressing these concerns to ensure a more professional and comprehensive presentation. The use of colloquial language has been eliminated, and the lists have been expanded and properly concluded without relying on "etc".

Furthermore, we understand your concern regarding the novelty of our research. We acknowledge the need for innovative contributions to the field. As such, we have revised the introduction and discussion sections to emphasize the novel aspects of our study, highlighting the unique perspectives and findings that distinguish our work from existing literature.

Once again, we appreciate your valuable feedback and assure you that we have taken it into consideration during the revision process. We hope that the revised manuscript now meets the expected standards for publication.

Reviewer 4 Report

The authors have answered all my questions. 

Minor editing of English language is reuired.

Author Response

Thank you for reviewing our manuscript. We appreciate your feedback and have considered your comments carefully.

We apologize for any English errors or problems in our submissions. Wherever possible, we will revise the manuscript to address these issues and ensure language is clear, concise, and of expected standards.

Once we have completed the necessary edits, we will resubmit the revised manuscript for your review.

Thanks again for your valuable comments.
